Testing the individual and social learning abilities of task-naïve captive chimpanzees (Pan troglodytes sp.) in a nut-cracking task

Neadle Damien damienneadle@outlook.com 1 2
Bandini Elisa 3
Tennie Claudio 3
1 School of Psychology, College of Life and Environmental Sciences, University of Birmingham , Birmingham , United Kingdom
2 Department of Psychology, School of Social Sciences, Faculty of Business, Law and Social Sciences, Birmingham City University , Birmingham , United Kingdom
3 Department of Early Prehistory and Quaternary Ecology, Eberhard-Karls-Universität Tübingen , Tübingen , Germany
Hopper Lydia
Electronic publication date: 2020 Mar 10
Publication date: 2020
Volume: 8
Electronic Location ID: e8734
Received 2019 Oct 11; Accepted 2020 Feb 11
Copyright: ©2020 Neadle et al.
Copyright year: 2020
Copyright holder: Neadle et al.
License: This is an open access article distributed under the terms of the Creative Commons Attribution License, which permits unrestricted use, distribution, reproduction and adaptation in any medium and for any purpose provided that it is properly attributed. For attribution, the original author(s), title, publication source (PeerJ) and either DOI or URL of the article must be cited.
License URL: https://creativecommons.org/licenses/by/4.0/

Keywords: Social learning, Chimpanzee, Individual learning, Sensitive learning period, Nut-cracking, Zone of latent solutions, Culture, Tool use, Copying, Pan troglodytes

Funding: Economic and Social Research Council Institutional Strategy of the University of Tübingen European Research Council under the European Union’s Horizon 2020 Programme H2020-EU.1.1 ERC grant Damien Neadle has received funding from the Economic and Social Research Council, under a full PhD Studentship ES/J50001X/1. Elisa Bandini and Claudio Tennie are supported by the Institutional Strategy of the University of Tübingen (Deutsche Forschungsgemeinschaft, ZUK 63). Claudio Tennie has also received funding from the European Research Council under the European Union’s Horizon 2020 Programme (H2020-EU.1.1.) / ERC grant agreement No. 714658. The funders had no role in study design, data collection and analysis, decision to publish, or preparation of the manuscript.

==============================
Nut-cracking is often cited as one of the most complex behaviours observed in wild chimpanzees. However, the cognitive mechanisms behind its acquisition are still debated. The current null hypothesis is that the form of nut-cracking behaviour relies on variants of social learning, with some researchers arguing, more precisely, that copying variants of social learning mechanisms are necessary. However, to date, very few experiments have directly investigated the potentially sufficient role of individual learning in explaining the behavioural form of nut-cracking. Despite this, the available data provides some evidence for the spontaneous acquisition of nut-cracking by chimpanzees; later group acquisition was then found to be at least facilitated by (unspecified) variants of social learning. The latter findings are in line with both suggested hypotheses, i.e., that copying social learning is required and that other (non-copying) social learning mechanisms are at play. Here we present the first study which focused (initially) on the role of individual learning for the acquisition of the nut-cracking behavioural form in chimpanzees. We tested task-naïve chimpanzees (N = 13) with an extended baseline condition to examine whether the behaviour would emerge spontaneously. After the baseline condition (which was unsuccessful), we tested for the role of social learning by providing social information in a step-wise fashion, culminating in a full action demonstration of nut-cracking by a human demonstrator (this last condition made it possible for the observers to copy all actions underlying the behaviour). Despite the opportunities to individually and/or socially learn nut-cracking, none of the chimpanzees tested here cracked nuts using tools in any of the conditions in our study; thus, providing no conclusive evidence for either competing hypothesis. We conclude that this failure was the product of an interplay of factors, including behavioural conservatism and the existence of a potential sensitive learning period for nut-cracking in chimpanzees. The possibility remains that nut-cracking is a behaviour that chimpanzees can individually learn. However, this behaviour might only be acquired when chimpanzees are still inside their sensitive learning period, and when ecological and developmental conditions allow for it. The possibility remains that nut-cracking is an example of a culture dependent trait in non-human great apes. Recommendations for future research projects to address this question are considered.

Introduction

Humans have created a unique niche within the animal kingdom, one that (most likely) relies on an ability to transfer knowledge between and within generations, allowing our species to inhabit almost every environment on the planet. However, modern industrialised human society is so far removed from that of our ancestors, that it is difficult to understand how our cultural ability—or our cultural niche (Odling-Smee, Laland & Feldman, 2003)—evolved. Therefore, the closest extant relatives of the hominin clade, non-human great apes (hereafter: apes), are often used as a phylogenetic proxy to help shed light on our own evolution.

Modern human culture may be unique in the animal kingdom (Tomasello, 1998)—at least in the technological domain (Tennie, Caldwell & Dean, 2018). Thus, to allow for the study of culture across species, a more minimal (or soft) definition of culture is required. Here we follow the terminology suggested recently by Neadle, Allritz & Tennie (2017), in which a cultural trait is any behaviour that is at least influenced (including merely its frequency being facilitated or catalysed; Tennie, Call & Tomasello, 2010) by social learning. Within this definition of culture, the social learning mechanisms at play can be any of the many proposed variants (for an overview, see Whiten et al., 2004). This minimal definition of culture allows for the identification of cultures that involve a range of social learning variants, including ones that rely on the copying of behaviour directly (often called imitation, see, e.g., Whiten & Ham, 1992). Crucially, the minimal definition of culture also encompasses examples of behavioural forms that can emerge without requiring social learning. Instead, behaviours that rely on (i.e., cannot occur in the absence of) copying variants of social learning are ‘culture dependent traits’ (henceforth CDTs; see Reindl et al., 2017; Tennie, Caldwell & Dean, 2018). Some animal behaviours and artefacts may be culture dependent (e.g., whale song is a candidate CDT; Tennie, Caldwell & Dean, 2018), but whether any animal tool use qualifies as CDT is a matter of considerable debate (Galef, 1992; Kendal, 2008; Tennie, Call & Tomasello, 2009). In the human case, technology has often evolved to the point that no naïve individual could reinnovate the behaviour on their own within their lifetime (Galef, 1992; Tomasello, Kruger & Ratner, 1993), making many modern human traits CDTs and indeed examples of cumulative culture (Boyd & Richerson, 1996). For a more in-depth discussion of these, and other, terms we refer the reader to a glossary of terms in the supplementary materials provided.

Amongst non-human animals (hereafter: animals), chimpanzees (Pan troglodytes) are, for now, the ‘most cultural’ species—at least in terms of (known) numbers of cultural traits (Whiten et al., 1999)—where a mere increase in sheer number of cultural traits is known as accumulation (Dean et al., 2014). However, accumulation (numbers of traits) should not be confused with cumulation, i.e., the cultural change of the traits themselves along transmission pertaining to cumulative culture (Dean et al., 2014). Only the latter is the result of the ‘ratchet effect’ (Tomasello, Kruger & Ratner, 1993), which underlies cumulative culture (Boyd & Richerson, 1996) and is responsible for CDTs (Reindl et al., 2017), which are the product of cumulative culture. Whilst chimpanzees’ number of cultural traits is impressive, for those concerned with human cultural evolution, the presence or absence of CDTs in chimpanzees is of particular interest.

Regarding chimpanzee culture, nut-cracking is considered one of the most complex behaviours expressed by any wild apes. Complexity can refer to the number of parts within a final artefact/behaviour (techno-units; Oswalt, 1976), the goals and sub goals of an action (Read & Andersson, 2019), the manual dexterity of an action (Foucart et al., 2005) and the number of “rules” necessary to describe the behaviour (Sirianni, Mundry & Boesch, 2015), amongst other metrics (see Vaesen & Houkes, 2017 for further discussion of complexity). Nut-cracking requires a high level of dexterity (Foucart et al., 2005) and involves several tools in various steps that need to be followed in a specific sequential order to produce the desired effect (Biro et al., 2003; Boesch et al., 2019; Inoue-Nakamura & Matsuzawa, 1997; Read & Andersson, 2019); thus, it can be considered a complex behavioural form. Furthermore, nut-cracking is rare across wild communities, (so far) only being documented in two geographically separate populations: two communities in West Africa (Bossou, Guinea and Taï Forest, Côte d’Ivoire Whiten et al., 2001) and one in Ebo Forest, Cameroon (Morgan & Abwe, 2006; although note that these data are based on indirect evidence and should be treated with some degree of caution).

The number of steps alongside the manual dexterity and use of multiple objects required for this behaviour suggests that nut-cracking is most likely a complex behaviour for chimpanzees. The basic behavioural form of nut-cracking consists of the following four sequential steps, though note that other steps might occur:

1. Place nut on anvil;

2. Pick up hammer (unless already picked up);

3. Lift hammer up;

4. Drop/push the hammer onto nut (all may be repeated).

Perhaps due to this apparent complexity, nut-cracking is often assumed to be culturally transmitted (Boesch & Boesch-Achermann, 2000; Lycett, Collard & McGrew, 2007; Lycett, Collard & McGrew, 2010), with some researchers arguing that action copying (or imitation) must be the mechanism responsible for its acquisition. For example, Boesch (1996) claims that chimpanzees learn how to crack nuts “by individual and social learning, including imitation” (Boesch, 1996, p. 418, emphasis added). Biro et al. (2003, p. 220) further argue that when nut-cracking “infant chimpanzees are driven not by a motivation for food but to produce a copy of the mother’s actions” (emphasis added). More generally, others agree, claiming that nut-cracking (alongside other chimpanzee traits) is difficult to explain “by social learning processes simpler than imitation” (Whiten et al., 1999, p. 685). More recently, similar claims have been made that chimpanzees rely on mother to infant “teaching” to acquire the skills required to crack nuts at a rate consistent with that of others within their community (Boesch et al., 2019). Some have further argued that young wild chimpanzees engage with this process during a so-called ‘sensitive learning period’ between the ages of approx. 3.5 years and 10 years (Inoue-Nakamura & Matsuzawa, 1997; Matsuzawa, 1994; Biro et al., 2003).

However, other research has suggested that dispersing primates, outside the estimated sensitive learning period, can still engage with and adopt behaviours—perhaps in keeping with their new groups. Most relevant here, Luncz, Mundry & Boesch (2012) describe how dispersed female chimpanzees adapt their hammer choice during nut-cracking to conform to that of their new group. The dispersed females were beyond the age of sexual maturity, and so outside their sensitive learning period. The fact that these individuals can modify their behaviour in this way, suggests that the possibility remains for individuals outside of the sensitive learning period to adopt the full behavioural form. This said, these findings are not evidence of the behavioural form of nut-cracking emerging for the first time, instead are evidence of behaviours being adapted and therefore whilst interesting might not represent a strong argument against the notion of a sensitive learning period.

In conclusion, the behaviours underpinning nut-cracking have been argued to require social learning (in particular action copying and/or unspecified variants of teaching). Therefore, it has been assumed that nut-cracking is outside of naïve chimpanzees’ individual learning abilities, which would make nut-cracking a CDT (sensu Reindl et al., 2017). This is a clear claim that can be tested. If nut-cracking requires social learning (if it is indeed a CDT), it should re-appear when a naïve chimpanzee has access to a model nut cracker to observe, thus suggesting that social learning is required for nut-cracking to occur. However, should we fail to identify nut-cracking, even at the end of this study, this should only be considered as an indication that it might not be a CDT rather than concrete evidence for it. Such a result would need replicating in another population before confident assumptions can be made from the data. Indeed, it is possible that environmental, social or individual factors might influence the likelihood of expression (sensu Tennie, Call & Tomasello, 2009). However, if nut-cracking does occur in a baseline condition in this naïve population, this would constitute evidence that naïve chimpanzees have the capacity to reinnovate nut-cracking in the absence of social learning (i.e., it is within the species’ ZLS).

In one formulation, the Zone of Latent Solutions (ZLS) hypothesis (Tennie, Call & Tomasello, 2009) posits that all non-human great ape ‘cultural’ behaviours can be reinnovated (defined by Bandini & Tennie, 2017) by naïve apes. This specific case has been termed as the ‘ZLS-Only’ hypothesis (Reindl, Bandini & Tennie, 2018). In line with this, Hayashi, Mizuno & Matsuzawa (2005) suggested that nut-cracking could potentially be individually reinnovated by chimpanzees. Some field reports support these views; for example, a report of nut-cracking in Cameroon (Morgan & Abwe, 2006) passes the ‘information barrier’ of the N’Zo-Sassandra River (McGrew et al., 1997). This report can be regarded as the outcome of a natural quasi-latent solution test (sensu Bandini & Tennie, 2018), as this pattern strongly suggests that that nut-cracking was (re-)innovated in two, culturally separate, wild communities (Tennie, Call & Tomasello, 2009, p. 2406). Though, again, these results should be considered in the light of the fact that they are supported by second hand reports, sounds in the forest and finding of tools.

If all underlying steps of the nut-cracking behaviour are also found to be reinnovated by a naïve, captive, chimpanzee in a culturally separate “island” of individuals (Tennie et al., 2016; Tomasello, 1999) then the behaviour would (by definition) cease to be a putative example of an animal CDT. This would support the ZLS hypothesis and would suggest that chimpanzees are capable, in principle, of individually learning the basic behaviour form underpinning nut-cracking; demonstrating that social learning is not required for this to occur. Importantly, it should be noted that social learning is likely to play a role in the process of chimpanzees understanding that nuts are a food source and, in addition, that they can be considered a ‘safe’ food (Hopper et al., 2011). Although this process is important for the frequency of nut-cracking within and across populations, our study is concerned with the mechanisms underlying tool-use aspect of the behavioural form of nut-cracking.

Therefore, here we test two competing hypotheses: chimpanzee nut-cracking as a culture dependent trait (the “CDT hypothesis”) versus chimpanzee nut-cracking as a behaviour that can be individually learned (re-innovated; Bandini & Tennie, 2017), but whose expression may nevertheless be facilitated by non-copying variants of social learning (the “ZLS hypothesis”, compare Tennie, Call & Tomasello, 2009; in press). Simply, the ZLS hypothesis posits that nut-cracking should emerge in a ‘baseline’ condition, i.e., without requiring social learning. Contrastingly, the CDT hypothesis argues that copying variants of social learning are necessary for the emergence of nut-cracking in a naïve sample.

Thus far, various chimpanzee behavioural traits, previously assumed to be culture dependent, have been reinnovated by naïve, captive subjects in latent solution tests (Bandini & Tennie, 2017; Bandini & Tennie, 2019; Menzel et al., 2013; Motes-Rodrigo et al., 2019; Neadle, Allritz & Tennie, 2017; Tennie, Call & Tomasello, 2009; Tennie et al., 2008). These behaviours (‘latent solutions’; Tennie, Call & Tomasello, 2009), were reinnovated without requiring any observation (or teaching). This does not, however, mean that social learning does not play any role in the innovation likelihood of these behavioural forms. Indeed, several variants of non-copying social learning (the specific mechanism was not directly tested in the studies mentioned above) greatly facilitate the innovation likelihood of the behaviour in both captive and wild chimpanzees (therefore affecting the observed frequencies of behaviours within and across populations; Tennie, Hopper & van Schaik, in press; Bandini & Tennie, 2017; Bandini & Tennie, 2019).

In the current study, we tested both the CDT and the ZLS hypotheses predictions for nut-cracking. In 2010, Tennie et al. hypothesised that nut-cracking would be within the chimpanzee ZLS, but that it may simply have a relatively low baseline probability of reinnovation (i.e., it is at the very edge of the chimpanzee ZLS). We were able to test both hypothesis by applying the extended latent solutions testing methodology (first described in Bandini & Tennie, 2018). This method first starts by testing for the reinnovation of the target behavioural form (here nut-cracking) in completely naïve chimpanzees (we ensured naivety by asking keepers of the animals’ previous experiences of the behaviour)—thus testing the ZLS hypothesis. If the behaviour does not appear in this baseline, subjects are then provided with incremental levels of social learning information. The particular methodology followed in this study allows for the examination of the role of individual learning (initial baseline test), then subsequently for end-state emulation, goal emulation and finally action copying (imitation) in the emergence of the target trait. By testing these competing hypotheses it is possible to determine whether the reliance on specific variants of social learning is restricted to our own species or whether it is also common to other apes. Should the former be supported by the present study, we would consider that comparisons between ape and modern human cultures should be largely suspended, until evidence of their similarity in using copying mechanisms is established. If the latter were true, it would suggest that the capacity for the resulting type of culture (cumulative culture, leading to CDTs) was shared by the last common ancestor between modern apes and humans.

Materials & Methods

Subjects

The subjects were 13 chimpanzees (Mage = 31.08; SD = 1; female = 9, male = 4; See Table 1; Pan troglodytes sp.). All subjects lived within a single group and comprised the entirety of that group, except for one individual (C13), which, due to group transfers within the zoological institution throughout the duration of this study, was introduced into the group before the start of the second condition (therefore C13 did not participate in the baseline condition). Subjects were provided with scatter feeds, consisting primarily of vegetables with some fruit in the morning (approx. 10am) and again in the afternoon (approx. 3 pm). The subjects were housed in two enclosures throughout the course of the study; between June 2017 and April 2018 subjects were housed in the “conversion” enclosure, from April 2018 until the end of the study subjects were housed in the “Eden” enclosure. Both enclosures consisted of two indoor areas and an outdoor area (two smaller areas in the case of conversion), with separate management areas (away from the observation of visitors). Subjects could be observed through glass panes in all public areas and mesh in management areas, observations used in this study were obtained from both. Within the main enclosures, subjects had access to enrichment devices, such as climbing frames/ropes, hanging feeders and nesting baskets. Other enrichment devices are regularly provided by keepers.

Table 1 Subject demographic information.

Note that subject names are anonymised for the purpose of the study—these codes were kept consistent throughout the study. Subject C13 is displayed in italics as she was only included in the study after the baseline condition.

ID	Sub-Species	DoB	Age (first testing day)	Sex	Rearing	
C1	Hybrid	30/04/1976	41	Female	Hand	
C2	Hybrid	09/06/1982	35	Male	Hand	
C3	Hybrid	25/10/1986	31	Male	Hand	
C4	Hybrid	18/08/1990	27	Female	Hand	
C5	Hybrid	28/12/1990	26	Male	Hand	
C6	Hybrid	10/08/2007	10	Female	Parent	
C7	Hybrid	25/05/1995	22	Female	Hand	
C8	Schweinfurthii	17/06/1977	40	Female	Undetermined	
C9	Verus	20/02/1988	29	Female	Hand	
C10	Verus	01/01/1965	52	Female	Undetermined	
C11	Verus	14/12/1971	45	Female	Undetermined	
C12	Verus	05/12/2003	13	Male	Parent	
C13	Hybrid	27/12/1982	34	Female	Parent	

Prior experience questionnaire

In order to exclude any possible influence of social learning on the results of this study, keepers filled out questionnaires and were interviewed (designed and distributed by EB at the zoological institution) about prior tool use behaviour. The use of this questionnaire was approved by the University of Birmingham STEM ethical review committee (ERN_17-1729). The questionnaire asks keepers to provide details on behaviours relating to “Using one object to bang on, or hit, another: usually, this means the use of a hard object to bang on or hit another, often hard, object. This may be with the aim to crack or break open the latter object, or to remove a substrate. Here, we are interested in any hammer-like behaviours, regardless of the objects involved”. This definition encompasses nut-cracking and similar actions, such as hammering behaviours. No instances of nut-cracking were reported in the questionnaire; however, a keeper described how one individual (C6; female; age 9) used a stone to tap on the glass of the outdoor enclosure. All but one keeper reported that the chimpanzees were frequently witnessed using their teeth to crack nuts, although they have never been provided with shelled macadamia nuts.

Ethical statement

All participation in this study was voluntary, and subjects were allowed to leave the testing area at any point throughout the session. Subjects’ usual feeding and cleaning routines were followed, minimising disruption to the animals. The experimental phase of this study was ethically reviewed and approved by the University of Birmingham Animal Welfare and Ethical Review Body (UOB 31213) and by Twycross Zoo Research Committee (TZR-2017- 013), following guidelines provided by SSSMZP, EAZA, BIAZA, WAZA on animal welfare and research in zoological institutions; this study also received a letter of support from BIAZA. This study adhered to legal requirements of the UK, where the research was carried out, and adhered to the ASP principles for the Ethical Treatment of Primates.

Motivation tests

This phase took place between 13th June 2017 and 27th September. Prior to starting experimental testing, it was important to ensure that the subjects were sufficiently interested and motivated to access the novel food reward (macadamia nuts) used in this study. To motivate the chimpanzees to try the nut kernels when first presented, the first stage involved a trusted individual (a keeper that has worked with the subjects for more than five years) first eating a different familiar food in front of the subjects (here we used dried raisins and berries). The keeper attracted a subject’s attention by calling their name, and then ate a single item of the familiar food (i.e., one raisin) in view of them. This process was repeated until each individual had observed the consumption in a group context. The subjects were then provided with the same food and required to eat it before moving onto the next step. As this food was familiar, this occurred in every case. The next stage was to introduce the novel food (macadamia nut kernels already without their shells). The same keeper ate a single macadamia kernel in the same way as with the familiar foods. Again, each individual was given a demonstration (sometimes groups of individuals could watch together as subjects were not separated during this part of testing). Once each individual had observed the consumption of the nuts at least once, they were provided with a macadamia kernel, again within a group context. This process was designed to increase the likelihood that the subjects would consume the novel food, as prior research has shown that captive chimpanzees can vary substantially in their acceptance of novel food sources (Visalberghi et al., 2002). Despite the neophobia reported by Visalberghi et al. (2002), we chose to replicate their ‘trusted’ human demonstrator condition in an attempt to maximise the likelihood that the subjects would consume the macadamia nuts. In addition to this, the ‘motivation tests’ were used to ensure that the nuts were palatable to the subjects; therefore, should they have succeeded in cracking a nut, they would be sufficiently motivated to continue doing so. Equally, during the demonstration conditions (see below) the nuts provided might then serve as a suitable motivator to encourage the chimpanzees to attempt to reinnovate or copy the behaviour. We required at least half the chimpanzees to eat the novel nuts before starting testing. This was to ensure that the motivation testing did not go on for too long, as these tests were carried out within a group context. It was likely that lower ranking individuals would never be allowed access to the nut kernels.

Test conditions

Each trial was video recorded, starting when the subjects were given access to the testing apparatus. The study took place between the morning and afternoon feeds; this time was chosen as it complemented the daily routine of the keepers and animals whilst providing the maximum testing time possible. The timings changed once the chimpanzees moved enclosure as the keepers were able to provide the afternoon feed without needing to move the subjects outside the testing area. Average trial length before the move was 3 h (n = 8), after the move it was 5 h 41 min (n = 12). Overall, there was a total of 92 h and 18 min observation time (Mtrial length = 4 h 37 min). The experimenter (DN) was present throughout each trial.

This study used a stepwise design, where each condition (see Fig. 1) was followed by the next in the event that the behaviour was not expressed in the first condition after five trials. For example, the “End state” condition was only implemented in the event that the behaviour was not reinnovated in the “Baseline” condition. Testing ended once the subjects had received 5 trials with full action demonstrations.

Figure 1 Decision tree depiction of the result dependent conditions.

If, at any stage, evidence of the behaviour was encountered then testing would cease, and the resultant learning mechanism will be attributed to the emergence of the behaviour. Each condition is continued for five trials before moving onto the next condition.

In all of the conditions, behaviours were first live coded. If, during live coding, any attempts at nut-cracking were identified then these were checked against videos and then second coded for reliability analysis. Here we defined nut-cracking in terms of tool use, therefore, to qualify as nut-cracking, the subject needed to use an object as a hammer to attempt to break open the nut, whilst resting the nut on another hard surface (the anvil). Video recordings were focussed on the experimental hammer and anvil set up (described below), however DN was present at all times to observe any behaviours that might have occurred outside of the camera frame.

Materials

The same apparatus set up was used in all conditions, and any changes to these conditions are noted in the relevant section. To set up the apparatus, DN entered the outdoor enclosure and secured a large wooden log (50 cm tall × 40 cm approx. diameter; that would serve as an “anvil”) to an upright portion of the climbing frame (which had a horizontal crossbeam, to ensure that the anvil could not be removed; see Fig. 2). The anvil was secured to the upright climbing frame using two 1m long, 8 mm thick, PVC coated, steel rope passed through two (12 mm diameter) holes drilled through the anvil (located 14 and 34 of the way down the log). Both ends of the ropes had a loop (secured by five ‘clips’ at each point, ‘clips’ used two, 8 mm, nuts and bolts;tightened using an electric drill), which was too large to pass through the hole in the anvil, and a steel padlock attached the two ends. Two of these securing attachments were used as a failsafe measure (see Fig. 2).

Figure 2 Securing attachment of the hammer.

Note how there are several ‘clips’ to act as a failsafe.

A wooden “hammer” was also attached to this structure (wooden, rather than stone, hammers were chosen as they were more secure in their attachment to the rope). The hammer consisted of a 30 cm long × 15 cm diameter log—approx. weight 2.5 kg—with a 12 mm hole drilled through half way along (see Fig. 2). Hammer length was chosen based on the descriptions of hammers used in wild populations to crack coula nuts (20–80 cm long; Boesch & Boesch, 1983). Our diameter was chosen to be larger than these wild hammers (4–10 cm in the wild; Boesch & Boesch, 1983) in the interest of safety being more likely to remain attached to the securing attachment (see below). As a result of this increased diameter the hammers were slightly heavier than the majority of those used in wild populations (77% of which were less than 2 kg; however, our hammers were still within the 2–4 kg larger range described in wild populations; Boesch & Boesch, 1983).

The hammer was attached to the anvil’s own securing attachment by creating a looped end in another (1.5 m) length of the same steel rope; the loop was passed onto the top securing attachment (of the anvil) and the loose end was secured to the hammer (by passing the loose end through the drilled hole and then securing with another five clips). The hammer was then moved less than 1m from the anvil (see Fig. 3).

Figure 3 Hammer and anvil set up within the subjects’ enclosure.

Note, the two securing attachments are passed through separate holes within the anvil and the hammer is less than 1m from the anvil (this was ensured by the length of the securing attachment of the hammer to the anvil).

The keepers then scattered three macadamia nuts (in their shell) per individual (i.e., 3 nuts × 13 individuals = 39 nuts) throughout the enclosure, avoiding a 2 m radius around the hammer and anvil set-up. The macadamia nuts were distributed at the same time as a regular scatter feed—just prior to the subjects being released into the outdoor enclosure. The unshelled weight of the nuts (around 1 g average across 10 measurements) was taken from the chimpanzees’ usual allowance of nuts for the week (this was to maintain the dietary health of the subjects, at the testing institution’s request). Once the attachments had been checked by DN and at least one keeper, all humans exited the enclosure and the chimpanzees were allowed in the enclosure. Just prior to the chimpanzees being allowed access, video cameras (SONY HDR-CX330e), set at two points framing the apparatus (to better capture various angles), on tripods, were set to record. DN was also present to live code relevant behaviours (see Table 2) that occurred outside of the frame of the fixed cameras.

Table 2 Coding ethogram used during the live coding procedure.

This was added to throughout live coding as behaviours of interest were observed. This ethogram was provided to the second coder for reliability coding.

Behaviour	Description	
Place nut	The subject places one/several nuts on the surface of the anvil. This is also coded if the subject drops the nut onto the anvil. The nut may roll off the anvil after being “placed” this is acceptable as it is likely due to the nut’s shape and the angle of the anvil’s surface.	
Hold hammer	The subject picks up the hammer—with the nut on the anvil, by holding the wood itself or the securing attachment.	
Raise hammer	The subject lifts the hammer above the nut—this may be at/below/above head height for the subject.	
Drop hammer	The subject brings the hammer down onto the nut, which must be resting on the anvil. The hammer can be dropped or held in the hand the entire time. This behaviour can be repeated until the nut is cracked. The behaviour is coded each time the behaviour occurs—i.e., each time the nut is struck.	
Eat nut	The subject takes the kernel of the, now broken nut and eats it. Note, this must have followed cracking of the nut by the subject.	
Stamp	The subject uses their foot to stamp on the nut, which has been placed on the anvil.	
Throwa	The subject, whilst sitting on the anvil, throws the nut in any direction.	
Notes.

a Note that throw did not require any behaviours preceding it.

Baseline condition

This test condition took place between 15th October 2017 and 30th November 2017.

In order to examine whether the subjects would individually reinnovate the target nut-cracking behaviour, it was necessary to test subjects without providing any social information beforehand. All sessions began between 10 am and 12 noon, when keepers provided the chimpanzees’ scatter feed (mainly consisting of vegetables and fruit). All sessions were conducted in the “Outdoor 1” section of the enclosure (see Fig. 4); however, subjects had access to both indoor areas throughout the session.

Figure 4 Experimental set up for baseline condition in “conversion”.

Note, the same set up was used for the first two trials of the “end-state” condition, prior to the enclosure move (see below).

End state condition

This phase of the study was completed between 15th January 2018 and 18th May 2018. However, after the first two trials (15th January 2018 and 17th January 2018) the weather conditions at the testing institution became so harsh that the subjects would often refuse to leave the indoor enclosure. Thus, testing was paused until 14th May 2018, after which the final three trials were completed on the 14th, 16th & 18th May. Between testing in January and May subjects were moved from “Conversion” (their previous enclosure) to a new enclosure: “Eden” (see Fig. 5); subjects were therefore given one month after moving to the new enclosure to settle in before testing resumed.

Figure 5 Experimental set up for “.end-state” condition in “Eden”.

Note, subjects had access to the entirety of this enclosure throughout these trials, however, the outdoor section of the enclosure was still under construction.

In this condition, we placed three macadamia nuts, shells and kernels, which had been split in half (see Fig. 6) on top of the anvil (in the “Conversion” enclosure this was in “Outdoor 1” and in “Eden” this was in “Habitat 1”). This condition was designed to specifically trigger stimulus/local enhancement (defined as when an animal’s attention is drawn to an object/location as a result of some change in the environment). In this condition, we drew a subject’s attention to the anvil and hammer (and the nuts) by adding the cracked nuts on top of the anvil.

Figure 6 Macadamia nut placement and state for end-state condition.

(A) Macadamia nuts, sawn in half (with kernels left whole) for the end-state emulation condition. (B) Nuts placed atop the anvil as described in text.

This condition was carried-out as the chimpanzees failed to individually reinnovate the nut-cracking behaviour in the baseline condition and followed the exact same protocol as the individual learning condition, described above. During the design process the study originally included an extra condition between the “Baseline” and “End state” conditions, called “Local Enhancement”. In this condition it would have been made clear to the subjects that a kernel is inside the macadamia nut and therefore that it constitutes a food source by shaving half of the nut shell away to reveal the kernel inside (see Bandini & Tennie, 2018). However, some of the chimpanzees in this study cracked the shells of the macadamia nuts with their teeth and subsequently consumed the kernels (see Fig. 7), rendering this condition unnecessary. This was unexpected as Boesch & Boesch (1983) state that they never observed a wild chimpanzee cracking coula nuts with its teeth; macadamia nuts (as used in this study) have a break strain of between 1,800 and 4,000 N (Schüler et al., 2014), which is comparable to coula nuts and substantially less than required for panda nuts (Boesch et al., 2017).

Figure 7 Adult male chimpanzee (C5) cracking a macadamia nut with his teeth, then eating the kernel.

(A) C5 biting the nut in an attempt to break it; (B) C5 consuming the kernel from the now broken shell.

Ghost condition

This phase of the study was completed between 19th July 2018 and 10th August 2018. The ghost condition involved a significant increase in the level of social information provided to the subjects. In this condition, the hammer and anvil set-up were still present inside the enclosure, along with three macadamia nuts per individual (scattered throughout “Eden Habitat 2”) and a further three nuts (this time whole and uncracked nuts, inside the shell was provided) placed on top of the anvil. Additionally, a replica of the equipment inside the enclosure (i.e., a hammer and anvil set up) was placed outside the enclosure, visible through the mesh near the subjects’ sleeping area (see Figs. 8 and 9); DN was also present, standing to the left of the anvil.

Figure 8 Experimental set up for “Ghost” condition in “Eden”.

Note, subjects had access to the entirety of this enclosure throughout these trials including the outdoor enclosure.

Figure 9 Experimental set up of “Ghost Condition” apparatus.

Note, the hammer is suspended by fishing line, and a single nut is in the centre of the anvil.

A reel of clear fishing line (0.65 mm diameter; 18 kg break strain) was attached to the hammer and passed through a section of mesh, allowing the hammer to be raised (between 80 and 50 cm) above the anvil, via a pulley-like system (see Fig. 9). A keeper steadied the hammer before dropping it onto the nut, thus increasing the likelihood of the hammer cracking the nut in the shortest possible time. Once the subject was clearly attending the apparatus (a subject’s attention was gained by calling their names), the hammer was dropped onto a nut (which was placed in a groove in the centre of the anvil), cracking the nut open—this did not always occur first time and may have required multiple attempts. A keeper then approached the anvil and gave the subject who watched the demonstration the cracked nut. The device was then rebaited with a new nut in the centre of the anvil. This procedure was repeated for a further 29 nuts (equalling a total of 30 demonstrations; with the exception of trial 4, where the line broke, meaning that the trial was halted after 17 demonstrations). Subjects had access to the testing apparatus during the course of the ghost trials. One camera was used to record the subjects’ interactions with the test apparatus, whilst the other was used to record subjects observing the ghost demonstrations; chest mounted GoPro (Hero5 Session) cameras were also used to record demonstrations and attention from the demonstrator’s perspective. In both this condition and the Full Action Demonstration condition, observing subjects were considered to be those in the enclosure immediately in front of the demonstration area (far left sleeping area in Fig. 8) oriented towards the apparatus/demonstrator (i.e., not with their back turned).

The ghost condition (inspired by Hopper et al., 2008) fulfils the primary stipulation of learning by emulation (Tomasello, Kruger & Ratner, 1993); i.e., the learner should not copy the motor patterns of the demonstrator. In this ghost condition, the motor patterns required for nut-cracking were not demonstrated, making it impossible for the chimpanzees to copy the actions (Heyes, 1994). Thus, if the behaviour were to occur following this condition, it could be inferred that the results of the actions were replicated rather than the actions themselves (Hopper, 2010; Whiten et al., 2004).

Full action demonstration condition (human demonstrator)

This phase of the study was completed between 16th August 2018 and 6th September 2018. The full action demonstration condition was the first one that allowed for the possibility of action copying. In this condition, DN was positioned outside the enclosure (in the same location as the ghost condition). An anvil was placed in the same location as in the ghost condition (see Fig. 8), with a hammer placed 1m from the anvil (both pieces of wood were identical to those in the subjects’ enclosure). The researcher then attracted a subject’s attention by calling their name and proceeded to crack a nut, on top of the anvil, using the hammer. Note, it was not possible to exclude the fact that multiple subjects may attend to the call of one individual—subjects attending to a demonstration were coded from videos. The experimenter used the hammer in a vertical manner, in the same orientation to the hammer in Fig. 9 (see Fig. 10), raising it to eye level and then hitting down onto the nut, resulting in the nut breaking open. The orientation of the hammer was used to attempt to control for hammer orientation between the ghost and full action demonstration trials. Once cracked, the kernel was provided to the subject by a keeper (see Fig. 10D) and the device rebaited with another nut. A total of 30 nuts were cracked using this procedure in each trial; a nut was not cracked until DN considered that the target subject was attending to the demonstration. A maximum of 30 nuts was used based on advice from keepers that not all subjects would attend to, or even approach, the demonstrations (therefore, for all subjects to observe, trials could have continued indefinitely which would have been unfeasible).

Figure 10 DN performing full nut-cracking action demonstrations.

(A) subjects’ attention was gained by calling their given name; (B) hammer was to eye-level and (C) brought down on the nut as many times as required until it cracked; (D) the cracked nut (both shell and kernel) are provided to the target subject by a keeper. Subject in this demonstration was the female (C9) to the left of DN—holding onto the mesh in A-C; keeper rolled the nut to C9 in D (hand feeding, even by keepers, is not permitted at the testing institution).

Coding/analysis of behaviours

Coding procedure

Trials were live coded using the ethogram in Table 2. Following live coding a formal coding procedure from video was followed. DN coded each trial in turn and a second coder (MT), naïve to the hypothesis of this study, second coded 100% of the behaviours identified (N = 31) along with an equal number of “dummy” clips where a subject was in the frame but DN did not identify a behaviour occurring to test for inter-rater reliability (acceptable Kappa would be 0.6; Cohen, 1968; calculted using R package “irr” v.0.84.1; Gamer et al., 2019). Note that the behaviours in Table 2 rely on the previous behaviour in order for them to be coded; e.g., if the subject picked up the hammer without first placing a nut on the anvil then the hammer behaviour would not be coded. This was to attempt to parse hammer centred play/exploration from attempts at nut-cracking.

Analyses

After a behaviour has been reinnovated, social facilitation cannot be excluded as a potential reason for the behaviour’s continued emergence in other group members (Bandini & Tennie, 2018; Tennie & Hedwig, 2009). Given an N of 1, it is not possible to perform inferential statistics on acquisition times or rates between individuals. However, descriptive statistics were used. All descriptive statistics were produced using R v.3.5.2 (R Core Team, 2013).

Results

Motivation test

During the motivation test, seven subjects (54% of sample; C3, C7, C9, C8, C5, C12 & C13) consumed at least one macadamia nut provided by the keeper, leading to the conclusion that macadamia nuts were/are palatable to most of subjects included in this study (although note that dominance hierarchies/individual personality characteristics may have interfered with certain individuals’ ability/motivation to access the nuts).

Reliability analysis

The results of a Cohen’s Kappa analysis revealed a strong level of agreement between coders (κ = .85, p < .001).

Attempts at nut-cracking

None of the individuals in this study attempted to crack open the nuts using a tool in any of the conditions described above. As there was never any evidence of nut-cracking, or approximations of it, all conditions were completed (as explained in the methods section).

Attempts recorded within the ethogram

The coding procedure identified the following behaviours from the ethogram: place (n = 26; first occurring during baseline condition trial 2 but distributed across baseline (n = 7), end state (n = 15) and ghost (n = 4) conditions), hold (n = 1; occurring during baseline condition trial 2), stamp (n = 2; occurring during baseline condition trial 2) and throw (n = 2; occurring during baseline condition trial 2). Recordings of “place” were identified in C5 (n = 7), C6 (n = 18) and C7 (n = 1) across all conditions apart from full demonstration. In only one instance did a “hold” event follow “place”, this concerned C6 during baseline condition trial 2; who was also the only individual to “stamp” on or “throw” the nuts. It is unclear whether throwing was an active effort to break the nut or simply an act of frustration/play as it did not appear that the throws were aimed at any hard surface, nor were there ever attempts to retrieve the nuts afterwards by the throwers.

Alternative techniques

Anecdotally, the majority of subjects (if not all) were witnessed, at least once, attempting to crack the nuts with their teeth (with some individuals succeeding; see Fig. 7). Male chimpanzees (n = 4) were the only individuals observed (by DN) successfully accessing the nut kernel using this method. The teeth cracking technique was first observed in the baseline condition and persisted throughout the study. These behaviours were not captured on the main videos as the cameras were facing the apparatus throughout the trial (to ensure that any attempts at using the apparatus to crack the nuts were captured), also some subjects were not visible throughout; therefore, any attempt to quantify these behaviours would be inaccurate as it would likely present only part of the actual series of events.

Observers in Ghost and Full Action Demonstration Conditions

Occasionally the identity of the observer could not be ascertained from video footage; in these cases, the individuals were not included in the calculations below. Furthermore, as participation in the study was voluntary, and subjects were free to approach and interact with the testing apparatus whenever they chose, not all subjects observed all the demonstrations provided. Some subjects (n = 2; C5 & C11) never observed the demonstrations in either condition; whilst other subjects never observed demonstrations in either the ghost (n = 3; C5, C13 & C11) or full demonstration (n = 4; C5, C11, C1, C10) conditions. Overall, 77% of subjects (n = 10) were coded as observers in the ghost condition and 69% of subjects (n = 9) were coded as observers the full demonstration condition; there was an average of 2.48 observers per ghost demonstration and 2.99 observers per full demonstration.

Discussion

We found no evidence of nut-cracking with a tool, or any approximation at this, at any point during the course of this study. Thus, our sample of 13 naïve chimpanzees failed to reinnovate or socially learn the behavioural form of nut-cracking. At first, it would seem our findings support the CDT hypothesis, in that nut-cracking behaviour was not reinnovated in our initial baseline condition. However, our study also consisted of various social learning test conditions—including one that demonstrated the necessary action patterns for nut-cracking to the chimpanzees. This condition allowed for the possibility of action copying being a requirement of the behaviour, as has recently been claimed (Estienne et al., 2019). Even so, nut-cracking was not acquired by the subjects. Therefore, our study does not provide conclusive evidence for either the CDT or the ZLS hypothesis. Below we discuss three possible explanations for our null result and the general disparity in studies of chimpanzee nut-cracking.

Conspecific models

The findings of this study raise the question as to why some chimpanzee populations in the wild regularly crack nuts (on average 270 nuts per day for as long as 2 h 15 min in Taï Forest; (Boesch & Boesch-Achermann, 2000) whereas captive chimpanzees (in this sample and others; Funk, 1985) seem to rather consistently fail to acquire the behaviour, even after demonstrations. A first possibility for the disparity between wild and captive data is that nut-cracking is indeed a CDT and requires the learner to imitate a conspecific demonstrator (Boesch, 1996). In this study we used human demonstrators, which may not have been considered ‘good’ enough models for the chimpanzees. Indeed, some research has shown that chimpanzees are more proficient social learners from conspecific models as compared to videos or human models (Hopper et al., 2015). However, in contrast, others have instead claimed that it is possible for chimpanzees to “learn” nut-cracking from human demonstrators (see findings of Ross et al., 2010) but also review of other related studies Tablle 3 (p. 230) of (Ross et al., 2010). Concurrently, other studies have found that, even with conspecific demonstrators, sometimes captive chimpanzees fail to acquire complex behaviours such as nut-cracking (Funk, 1985) or behaviours which strictly require imitation (Clay & Tennie, 2018; Tennie, Call & Tomasello, 2012; Tomasello et al., 1997). Although it might have been interesting to observe the chimpanzees’ reaction to a conspecific demonstrator in this study, we did not have the resources to train a chimpanzee to act as a demonstrator, but we encourage interested researchers who do have the resources to replicate this study, and include a conspecific demonstrator to observe whether this affects the findings presented here.

Behavioural flexibility

An alternative explanation for the fact that nut-cracking did not emerge in this study is that the chimpanzees were hindered by their lack of behavioural flexibility, a commonly recorded phenomenon in chimpanzees (e.g., Harrison & Whiten, 2018). The chimpanzees in the current study seemed to become fixated on one solution to open the nuts: i.e., the use of their teeth. The chimpanzees may have relied on this technique due to their pre-existing knowledge on how to crack softer-shelled nuts (such as peanuts and walnuts), which they are often provided during their feeds at the testing institution. These nuts are easily cracked open by apes using teeth (DN; personal observation, keeper reports and see also Visalberghi et al., 2008 for measurements on the required force for different types of nuts). The heuristic (Marsh, 2002) in this case may be that nuts (in general) can be opened with teeth—and indeed our macadamia nuts were no exception. Chimpanzees have been shown to be reluctant to display behavioural flexibility in abandoning a previously successful solution (see Harrison & Whiten, 2018; Hrubesch, Preuschoft & van Schaik, 2008; but see also Manrique, Völter & Call, 2013). Thus, it is possible that the first individual to successfully crack a nut with the use of teeth (see Fig. 7), facilitated this behaviour within the group and/or that other individuals independently converged on this method, and then the subjects were unable to innovate a new method, even if cracking the nuts with a tool would have been mechanistically easier/more efficient (this possibility is also in line with cultural founder effects; Tennie, Call & Tomasello, 2009).

Sensitive learning period

Based on the literature, the most likely explanation for the findings in this study is that the chimpanzees may have simply been outside of their sensitive learning period for nut-cracking. Previous studies on nut-cracking in wild chimpanzees have reported that before 3.5 years, juvenile chimpanzees are unable to express the full nut-cracking behavioural form (Inoue-Nakamura & Matsuzawa, 1997; Matsuzawa, 1994). However, juvenile chimpanzees (as young as 1.5 years old) that had been exposed to the materials required for nut-cracking at various ages/developmental stages were able to perform the basic actions of the behaviour (put, hold, hit and eat), but not combine them in the required order to perform the full nut-cracking behaviour (Inoue-Nakamura & Matsuzawa, 1997). Indeed, recent research in wild chimpanzees shows an exponential increase in nut-cracking between 5 and 6 years of age, though the first signs were observed in 3–4-year-old individuals (Estienne et al., 2019) in line with the concept of maturation (Corp & Byrne, 2002). This finding suggests a certain level of developmental prowess required to express nut-cracking, perhaps somewhere between maturation effects of the body and the brain.

In addition to this lower age limit for the acquisition of nut-cracking, there also appears to be an upper limit (more relevant for the current study). A 13 year longitudinal study by Biro et al. (2003) found that wild chimpanzees who did not learn the basic nut-cracking skills before five years old seemed unable to acquire the behaviour later on in adulthood (a similar case has been documented recently for stone tool-use in long-tailed macaques; Tan, 2017). The subjects tested in the current study were all outside of the hypothesised sensitive learning period for nut-cracking, as the youngest subject in our sample was already 10 years old at the time of testing. The youngest individual however was the only subject to display the “hold” behaviour (stage two of four) in the behavioural form of nut-cracking. Our findings, coupled with those described here suggest that a sensitive learning period may be a decisive factor for whether a chimpanzee will start to crack nuts or not (leaving open the question how this is learned, i.e., whether it is a CDT or a latent solution).

Given the fact that wild chimpanzees engage in an extended process of acquisition before expressing nut-cracking (Matsuzawa et al., 2008), we suggest that future work considers applying an even longer study time than the one employed here. It is possible that chimpanzees may then individually, or socially, learn the behaviour. The social learning oppertunities here provided comparatively more demonstrations than related studies (5 trials with 30 nuts per demonstration in our study versus 5 trials with 5 nuts per demonstration by Marshall-Pescini & Whiten, 2008). Even so, the social learning opportunities that we present here were fewer compared to the wild; in wild populations that express the behaviour, individuals have more and longer (and perhaps also more varied) opportunities to observe nut-cracking. Equally, given the potential importance of a sensitive learning period in explaining the emergence patterns of chimpanzee nut-cracking (discussed in this section) we suggest that the next logical test of this behaviour should aim to test younger chimpanzees between the ages of 3 and 10 years (Ross et al., 2010 suggest between ages 3–7); either way, these individuals should once again be selected from populations that have not been observed previously to crack nuts.

Though we used a within-subject design throughout our result-dependent design, we would reccomend that (wherever feasible) a between-subjects design be used in future tests (one group for each of the conditions). By doing the latter, it is possible to control for and measure the time of exposure required for chimpanzees to express nut-cracking, and it would exclude potential carry-over effects. However, this project would likely be an overly large undertaking for any one research group, so therefore may be better suited to large scale collaborative projects (e.g., the ManyPrimates project).

Conclusions

Although no chimpanzees in this study demonstrated nut-cracking using tools, two geographically separate populations in the wild have seemingly converged on the same method for cracking nuts using tools (West Africa; Whiten et al., 2001 and Cameroon; Morgan & Abwe, 2006). As these populations do not have access to each other, logically they must have independently reinnovated nut-cracking (Byrne, 2007). However, the data from Cameroon is based on procured nut cracking tools and auditory assumptions and so are an inference rather than direct observations. It is possible that these tools were actually from modern human nut-crackers and may not be considered evidence of chimpanzee nut-cracking in multiple populations. Thus, to date, there is only concrete evidence of one culturally independent wild population expressing nut-cracking. As the data from this study and wild data do not unequivocally support either the ZLS or the CDT hypothesis, both remain in contention.

In addition to the wild data and that presented here, one chimpanzee in an experimental study spontaneously reinnovated nut-cracking when provided with all the materials (Marshall-Pescini & Whiten, 2008). The individual (Mawa) acquired the nut-cracking behaviour seemingly without requiring any copying variants of social learning (Marshall-Pescini & Whiten, 2008). However, Marshall-Pescini & Whiten (2008) fail to note the importance of these findings, by assuming, based on the speed of acquisition, that “Mawa” had prior experience of nut-cracking. It is worthy of note however that unlike this study and that of Funk (1985), no claim was made about the naivety of the subjects. Indeed, Mawa was kept as a pet prior to residing at the sanctuary where Marshall-Pescini & Whiten (2008) carried-out their study. Mawa arrived at the sanctuary when he was approx. three years old with wounds from a rope where he was tied up (Ferdowsian et al., 2011). As a result of this potential enculturation, or at the very least deprivation, these data should be treated with caution; indeed, the generalisability of such individuals to wild chimpanzees is questionable (Henrich & Tennie, 2017).

Excavations of chimpanzee nut-cracking sites suggest that the basic behavioural form has remained constant for at least 4,000 years, and likely even longer (Mercader et al., 2007). Similarly, excavations of capuchin nut-cracking sites have demonstrated that their nut-cracking form has remained the same for 3,000 years, with only the tools (hammerstones) changing in shape over time (Falótico et al., 2019). Indeed, if the behavioural form of nut-cracking were being copied between individuals, we would expect to see some changes to its form over time due to copying error alone (see Eerkens & Lipo, 2005). Lastly, other extant species of primates also crack nuts using tools in the wild (long tailed macaques, Gumert & Malaivijitnond, 2013; capuchins, Ottoni & Mannu, 2001), and some have even been found to do so spontaneously in captivity without requiring social learning (e.g., nut-cracking is a latent solution in capuchins: Visalberghi, 1987). Whilst other primate species being able to spontaneously crack nuts is not evidence of a phylogenetic link; it does suggest that the possibility that nut-cracking is a latent solution in chimpanzees remains. Therefore, social learning may not be fully responsible for the emergence of nut-cracking in chimpanzees. We acknowledge that the chimpanzees in this study were captive and therefore are not subject to the same ecological pressures as their wild conspecifics; that is, they would have less ‘necessity’ to reinnovate the behaviour (Fox, Sitompul & van Schaik, 1999). Therefore, (parts of) this study could perhaps be replicated in a wild sample, naïve to nut-cracking.

Therefore, the results of this study do not support nut-cracking as the first evidence of a CDT in chimpanzees (see also Byrne, 2007), yet they also do not support nut-cracking as a latent solution in chimpanzees. Instead, we conclude that the behaviour may not have emerged here due to interplay of factors, including a certain level of behavioural conservatism and, crucially, the fact that all the subjects were already out of their sensitive learning periods for nut-cracking. We believe it is unlikely that our use of human demonstrators was the reason for the failure of all our subjects to express nut-cracking, given the results of previous studies, discussed above. Accordingly, we propose that future studies should adopt the methodology presented here, but test unenculturated infant/juvenile chimpanzees, naïve to nut-cracking and to opening nuts with their teeth, to remove the confounds of the sensitive learning periods and conservatism (ideally tested in isolation in order to increase effective sample size). Under these conditions, it is plausible that some naïve chimpanzees will reinnovate nut-cracking. Yet, on the other hand, given the extended trial-and-error learning process that young wild chimpanzees engage in (Matsuzawa et al., 2008) it is possible that under the relatively short term test conditions, the full form of nut-cracking may still fail to emerge spontaneously, although some of the pre-requisite steps to the behaviours may still develop.

So far, the current state of knowledge does not support the view that nut-cracking has to be reliant on social learning as it has potentially been reinnovated in two culturally distinct populations, therefore, it seems unlikely that it is a CDT. However, it is also possible that chimpanzees within their sensitive learning period may fail to individually acquire the skills required to crack nuts; in this case nut-cracking could not be considered a CDT. The data at hand suggest that the behavioural form of nut-cracking may only be acquired through an interplay of ecological and developmental factors, i.e., chimpanzees must be in a location with appropriate nuts and tool materials, during or before, their sensitive learning period. Therefore, it remains possible that nut-cracking is within the species level ZLS of chimpanzees. Despite this, not all individuals may realise this potential within their lifetime if they were not exposed to the required ecological conditions or individual prerequisites (note that these were termed by Tennie, Call & Tomasello (2009) as the “right” conditions that may be required). It is yet to be determined whether nut-cracking’s acquisition is best described as being due to, and requiring, social learning (culture-dependent) or is due to socially mediated reinnovation (latent solution). Further research should consider the importance of the ecological factors explored here in addressing this question.

Supplemental Information

Supplemental Information 1 R script used to produce the descriptive statistics in text

Click here for additional data file.

Supplemental Information 2 Attention data for Full Demonstration Condition

Click here for additional data file.

Supplemental Information 3 R script used to conduct Cohen’s Kappa analysis

Click here for additional data file.

Supplemental Information 4 Data used for Cohen’s Kappa calculations

Click here for additional data file.

Supplemental Information 5 Attention data for Ghost Condition

Click here for additional data file.

Supplemental Information 6 Glossary of Key Terms

Click here for additional data file.

We would like to acknowledge the support of Twycross Zoo throughout the process of this study; without their support in material acquisition and access to subjects, this study would not have been possible. In particular, we would like to thank Clare Ellis, Freisha Patel, Katie Waller, Kris Hern, the entire ape team and the staff at Wates for their support. In addition, we would like to thank Jackie Chappell, Sarah Beck, Susannah Thorpe and Josep Call for their helpful discussions regarding this study. We would also like to thank Lydia Hopper, Lydia Luncz and two other anonymous reviewers for helpful comments on an earlier version of this manuscript. Finally, we would like to thank Matthew Thompson for his help in completing second coding for reliability analyses.

Additional Information and Declarations

Competing Interests

Author Contributions

Animal Ethics

Data Availability

The authors declare there are no competing interests.

Damien Neadle conceived and designed the experiments, performed the experiments, analyzed the data, prepared figures and/or tables, authored or reviewed drafts of the paper, and approved the final draft.

Elisa Bandini and Claudio Tennie conceived and designed the experiments, authored or reviewed drafts of the paper, and approved the final draft.

The following information was supplied relating to ethical approvals (i.e., approving body and any reference numbers):

University of Birmingham Animal Welfare and Ethical Review Body committee (UOB 31213) and Twycross Zoo Research Committee (TZR-2017- 013) provided full approval for this research.

The following information was supplied regarding data availability:

All code and data required to replicate our data are available in the Supplemental Files.

Data are also available at Figshare: Neadle, Damien; Tennie, Claudio; Bandini, Elisa (2019): Chimpanzee Nut-Cracking PeerJ. figshare. Dataset. https://doi.org/10.6084/m9.figshare.9962756.v1.

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
