# Peer review of "Testing the individual and social learning abilities of task-naïve captive chimpanzees (Pan troglodytes sp.) in a nut-cracking task"

_PeerJ, doi:10.7717/peerj.8734_

## Round 0.1 · original submission · Minor Revisions

Thank you very much for your submission to PeerJ. I have now been fortunate to receive reviews of your article from three experts in your field. All three provided thoughtful and detailed reviews that I think will help you as you revise your article.

While all three reviewers provided favorable reviews of your article overall, they all had a number of comments and suggestions as to how you might enhance the clarity of your article, especially with regard to the theory you present and the motivation for your experiment. I agree with all their points and I encourage you to consider each of their points and provide detailed responses to their feedback as you prepare your revision for resubmission.

In addition to their comments, I have three editorial points that I want to highlight (some of which are points also raised by the reviewers):

1. When you resubmit your article, please ensure the text is double spaced - this makes it much easier to read and review your submission.

2. Throughout, please avoid the use of footnotes. While PeerJ can accommodate a limited number of short footnotes we encourage authors to omit them altogether. If the information is prudent, please simply include it in the body of your text, or, if it is not relevant, simply omit it.

3. I agree with the authors that you could provide a more detailed data set as your raw data provided with this article, especially in terms of any interactions with the task or nuts at other stages of the experiment.

I look forward to receiving your revised article.

·

Basic reporting

This paper is an interesting approach to an important and widely discussed research topic. In recent years, research has shifted from debating the existence of culture in great apes to trying to dissect its underlying mechanisms and potential similarities and differences to human culture. Those mechanisms underlying animal culture are still debated with controversial input from different scientific fields. I congratulate the authors for publishing negative results, which are mostly still absent in our research field, yet they can provide important information in itself as well as information to improve future research into similar topics. Field work with great apes is often limited in performing experiments and this study has addressed this with standardized experiment in a captive setting, building upon previous work on testing the occurrence of spontaneous behaviour that has been labelled cultural in wild animals.

Experimental design

The paper is well written, and the study was carefully designed. Similar tests have been done beofore, but the stepwise approach to test the hypothesis in this paper is adding a new and coherent level. However, I have some suggestions for improvement as well as some more serious concerns that I would like the authors to address before I can support the publication of this work in PeerJ.

Validity of the findings

The conclusions presented in the manuscript are not supported by the results. Neither social nor individual learning was found to trigger nut cracking using this setup, however the authors seem to favour the option where nut cracking can be learned individually. This “conclusion” needs to be toned down. Especially as in the introduction (Line 140) the authors state that if nut cracking is a CDT it should not re-appear. Listing the limitations of the study in the discussion is a very valuable contribution to future research using a similar set up. However, it seems that the authors then disregard the severity of these limitations and conclude in favour of the ZLS. In captive animals there is most likely a lack of motivation, these animals are well fed and safe and do not depend on cultural dependent traits for survival. This does not hold true for wild chimpanzees and needs to be discussed in the manuscript.
Nut cracking in wild animals takes years to acquire. Maybe the length of the task needs to be extended. The exposed time period might have been too short. If the conclusions are to assume an age sensitive learning phase, then testing chimpanzees in the predicted age within this learning phase would be critical. I would recommend emphasising this in the discussion part for other researchers to take this into account when replicating the study.

Additional comments

Recent research (Gumert et al. 2019) suggested that potentially some degree of genetic predisposition might help facilitate the learning of proficient tool use in primates. Until today only Western chimpanzees (Pan troglodytes verus) have been observed to use stone tools. It would be important if the authors could inform the reader of the subspecies of chimpanzees that were participating in the experiments. In case this is not known I suggest to at least discuss the possibility of the importance of the sub species.
Your introduction needs more detail. There is a body of literature that needs to be included into this work when discussing potential formative phases in primates. Research of migrating primates has repeatedly shown that that adult wild chimpanzees as well as vervet monkeys conform to behaviours of their new groups, demonstrating awareness of behavioural traits even past what the authors claim to be a formative phase. I would like to see this literature discussed or at least acknowledged in order to provide the reader with adequate information of the topic at hand (see work by Luncz et al. and van de Waal et al.) For further examples of behavioural awareness during migration in wild animals please see Aplin et al. and her work on birds.
Wild chimpanzees are very neophobe and learn what is eatable and what is not from their mothers during upbringing. Later in life they hardly ever explore new food sources as it can come with detrimental costs (which is different from a “formative phase” as they are still “culturally aware” and adjust to local behaviour). Therefore, nut cracking is not only a tool behaviour but also it needs to be learned that this food is harmless. Acquiring the knowledge of eatable food is already social learning and a crucial step in the CDT of nut cracking. This needs to be considered and cannot be disregarded as some prerequisite for the “actual” nut cracking behaviour of lifting a stone up in the air.
In the abstract as well as in the discussion the phrasing “at the very edge of individual learning” or “at the brink of the ZLS” needs to be changed. That is confusing to the reader and does not help the debate of social learning. Either a behaviour can be individually learned or not. If the authors intend to say that a behaviour can be learned but not to full proficiency, they need to make this clear in the text. However, if a behaviour cannot be learned to proficient levels, I would assume that it does not classify for the ZLS.
Nut cracking is also known in Liberia and other locations in the Republic of Guinea (see Kuehl et al. 2019 sub material for a complete table of the distribution of nut cracking in West Africa).
The report of nut cracking in Cameroon (Morgan and Abwe 2006) is based in secondary evidence of nut cracking (shells and stone tools). Chimpanzees in Cameroon have never been observed to nut crack and efforts to find further evidence has not been successful (also see Kuehl et al. 2019). The report of this behaviour in Cameroon therefore has been treated with caution recently as it might have well been carried out by humans. To build a strong argument on this study is disregarding the concerns experts have raised with this finding. The controversy of this paper needs to be discussed. Strong claims like that need to based on more evidence.

Minor issues:
Line 157: Tennie 2009 in press, what does that mean. In press usually does not have a publication year.
161-164: Please rephrase the hypothesis, and explain in more details what you are going to test. It is not easy to follow.
Methods: Please include more details of the setup of this study (zoo, sanctuary, feeding times, group composition)
Line 240: Age nine is not adult
Add the total number of hours tested. In the Abstract the authors state extensive learning opportunities for nut cracking are provided in this experiment. In the wild nut cracking takes years to acquire, please delete extensive.
Line 271-275: Please explain this further. Break down this sentence into smaller sections, is very long and contains multiple lines of information.
Line 280: typo: texting
Figure 2: we cannot see the anvil. Please provide a picture that shows the dimensions of the objects included.

Reviewer 2 ·

Basic reporting

The manuscript is well written in clear language. There are places in the introduction that feel slightly overwhelmed by terminology. I suggest a table with definitions might make it easier for the reader to keep on top of the terms. Terms included in the table could include cultural trait, CDT, cumulative culture, ZLS, accumulation of traits etc.

I do feel that the authors miss out some relevant literature on the development of nut cracking abilities in wild chimpanzees – especially with regard to the time scale of the development of this skill (see below for more discussion).

Raw data and code are provided online.

Experimental design

The methods were detailed well in text and using clear figures and I believe they are in enough detail to sufficiently replicate the study.

However, there are some issues with the experimental design that I feel should be addressed in the manuscript which I detail below:

Line 135, the authors state that: “If nut-cracking requires social learning (if it is indeed a CDT), it should re-appear when a naïve chimpanzee has access to a model nut cracker to observe.”

Although the authors go on to say that to test if it’s a CDT, testing for its re-innovation without social learning is necessary, this sentence still suggests that behaviours that require social learning should reappear if given social demonstrations. There could be many reasons why a behaviour reliant on social learning wouldn’t emerge after demonstration, including insufficient period of demonstration, social reasons such as rank, social network, and individual differences such as age, motivation, learning ability and many more. I would suggest rephrasing to avoid giving the impression that the lack of a behaviour after a demonstration is good evidence that it’s not a CDT.

The social opportunities presented by the authors to the captive chimpanzees are very different to those afforded to the chimpanzees that exhibit these nut-cracking behaviours in the wild. Whilst the effort to separate the different types of social learning (through stimulus enhancement and ghost condition) is commendable and interesting, these opportunities are extremely restricted when compared to their wild counterparts and I think this needs to be discussed.

The authors mention themselves in the discussion that (L518) “some chimpanzee populations in the wild regularly crack nuts (on average 270 nuts per day for as long as 2 hours 15 minutes in Taï Forest)”. You also mention very briefly (on line 625) the long trial and error learning required by wild chimps to crack nuts. So the number of social demonstrations provided to the learner must be many, many times more than the 30 used in the current study. I’m surprised therefore that the authors have not mentioned this before in the introduction or in relation to their own results as it seems to be extremely pertinent – if wild chimps take so long to exhibit this behaviour, whether through individual learning or action copying - why then did the authors decide that 5 trials of roughly 3 hours would be sufficient to test this behaviour at each stage?
The authors could mention here that the (young) chimps in Marshall-Pescini & Whiten (2008) learned to crack nuts in a very short time period – was this the justification for the short time period? I feel that there needs to be a lengthier discussion of this extended learning time in the wild in relation to the aims, results and limitations of this study.

Similarly, given the well-known literature on critical learning periods in nut-cracking in wild chimps (which the authors discuss in relation to why innovation was not seen), I feel that the authors should provide some rationale in the introduction for testing adult chimps in this context.

Validity of the findings

The nature of the study meant that the authors were unable to use inferential stats, but provide descriptive stats of their results. I think more details are required about the behaviours that did occur (particularly the "place" behaviours, see detailed comments below).

The study has interesting aims and I think the results would provide valuable information to other researchers in the field. However, I suggest that the authors need to be clearer in the manuscript about the limitations of the study, how this may have affected the results, and what they can conclude from the results.

Since the chimps did not exhibit nut cracking behaviour under any conditions, the authors conclude that the data did not provide conclusive evidence for either of their hypotheses. However, they also claim that (L628) “the findings of this study, and others, do not support the view that nut-cracking has to be reliant on social learning, i.e., it does not seem to be a CDT. Therefore, it remains possible that nut-cracking, while inside the chimpanzees’ ZLS, is at the very brink of it.”
The language used here seems to suggest that the authors find the results to point more towards the ZSL hypothesis than the CDT hypothesis. However, I'm not convinced that this is warranted from the current results.
The main results, as far as I can see, are that these adult, naïve chimps did not individually innovate nut-cracking, nor did they copy nut-cracking from a human demonstrator (or by using the other tested social learning forms) during a relatively short time period. There is support for neither hypothesis and I think the authors need to be clearer about this in some areas.

Additional comments

Please consider formatting your ms in double space for future reviews – it makes it a lot easier to read.

Abstract:
I suggest making clear in the abstract that it is a human doing the full demo as this is quite an important point that the reader probably wants to know at the earliest opportunity.

You say the chimps were given “extensive opportunities” to learn the behaviour. Were they really that extensive? Compared to previous experiments or wild conditions?

L316 – Is this hammer a similar size/weight/shape to hammers used in wild populations? If so, would be good to state. If not, also state why it differed and whether this may have any effect.

L375 – some chimps cracked the nuts with their teeth pretty early on (you later discuss this with regard to behavioural conservatism). Does this ever occur in the wild or are the available nuts in their habitats all too hard to achieve this?

L475 - It's unclear from the results when these behaviours occurred. Were they during the baseline phase or the social learning phases (if so, which ones?) - this seems especially interesting for the "place" behaviours (see below)

L477 - You recorded a number of "places". Why would the chimps place the nut on the anvil so many times? Did they similarly place nuts on other surfaces around the enclosure? During which phase did these behaviours occur? It's hard to tell without more information, but it seems possible that this placing behaviour could potentially be the beginning of the nut-cracking sequence. If they were given more time to individually learn/view more demos, would this placing behaviour then be followed by the other stages of nut-cracking? I think the authors need to give more details about the context of these behaviours.

Reviewer 3 ·

Basic reporting

Overall, this manuscript clearly explains the experimental design and results of the research, and does so while citing a wide range of the relevant literature. The extensive data collection phase consisted of baseline, end-state, ghost, and full action demonstration conditions wherein 13 captive adult chimpanzees were afforded the raw materials (hammer, anvil, nuts) for nut-cracking. Although none of the chimpanzees used the tools to crack the supplied nuts (some used their teeth), a number of interesting conclusions regarding, e.g., sensitive periods, are posited. Several improvements to the manuscript are desirable before it is suitable for publication.

First, the Introduction lacks focus. The descriptions of the competing hypotheses (nut-cracking as a culture-dependent trait vs. latent solution) are separated from each other by several paragraphs, making it difficult to properly contrast them in meaning & implication. The paragraphs beginning on Lines 93 (complexity) and 166 (latent solutions tests) could be cut down significantly or even eliminated.

Related to this, it was difficult to follow what the requirements of culture-dependent traits are. It seems that social learning is required (line 74), and that change of a trait via transmission (lines 86-89) is also required. Is this the case? Or would a trait that requires social learning but is unchanged after transmission qualify as a CDT as well?

Second, why is it important to determine whether nut cracking (and other potentially cultural behaviours) are CDTs versus individually acquired/innovated?

Third, it is correct to state in the Abstract (lines 50-52) and Conclusion that the results obtained here do not rule out the possibility that tool-based nut cracking is a behaviour that could be re-innovated in the absence of social learning. It is, I think, important to acknowledge in both places that the results also do not rule out the possibility that nut cracking is a behaviour which requires social learning/a CDT (as is done in the Discussion).

The shared data file contains information about the individuals observing full action nut-cracking demonstrations. Can the data for the ghost demonstrations also be shared? It is difficult to link an individual chimpanzee listed by name in the data file to their number from Table 1.

Experimental design

Useful experimental design, clearly described. A few questions for clarification:

-How many types of familiar foods were given in motivation trials? Just the 2 listed?

-What does “preference testing” (line 273) refer to?

-How would the not-carried-out “local enhancement” condition have differed from the end-state condition? Would all of the scattered nuts (not just those on the anvil) be half-shelled?

Validity of the findings

Statistics were not performed on the nut-cracking data, as in the end none of the chimpanzees used the tools to crack the nuts. However, the authors provided a good amount of detail regarding inter-rater reliability, trial duration, the number of chimpanzees who observed/did not observe various demonstrations, and other events.

Some important information that should be included in the Results is the condition(s) in which the relevant behaviours (place, hold, stamp, and throw, lines 476-482) occurred, in either the text or a table.

Would it be possible to report information about the number of nuts that ended up being consumed in the trials? i.e., was it possible to retrieve any unopened/opened nut shells from the enclosure(s) after the trials? That could give a sense of how many of the nuts were opened by mouth and consumed.

An additional useful piece of information would be whether the end-state condition had (one of) the desired effect of increased chimpanzee interest in the anvil area compared to the baseline condition, if this is straightforward to extract from the existing data.

Additional comments

If the authors so desire, the title could be re-worded to “…task-naïve captive adult chimpanzees…” for readability.

In the Abstract, do lines 34-39 refer to a particular study? It would be helpful to cite this here.

In the Abstract, it is noted that “none of the chimpanzees tested here developed the behaviour”. However, some of the individuals were able to open the nuts with their teeth; therefore, it would be useful to specify that none of them developed the method of nut-cracking *which required the use of tools*. Same for Conclusions, line 586.

Check spelling of author name in reference Odling-Smee et al. 2003.

In the Introduction (line 153-156), the contingency should go the other way.

Regarding the minimal reported “hammering” behaviour in this group of chimpanzees, I am curious if they are ever given whole coconuts. In other zoos, these are given as enrichment and are opened by banging/dropping/throwing. This could be a relevant antecedent behaviour.

Line 278 complimented -> complemented

Line 280 texting -> testing

In Figure 2 (and 3), it is not immediately clear which thing is the anvil, and which is the hammer. Labelling would be helpful.

What are the “relevant behaviours” referred to in Line 335? The ones listed in Table 2, or others?

Throughout: check subjects’ vs subject’s vs subjects (and individuals’ vs individual’s) (lines 364, 403, 425, 466)

Table 2: was “place” required before “throw” for it to count? If not, please note this where coding is described (line 448-450) or in a footnote to the table.

Why was C13 present for the motivation test (before the baseline condition) but not during the baseline condition? Were there motivation tests before every condition?

Please clarify what “(n=13)” in lines 500-501 refers to.

I am curious whether, in the behavioural flexibility account for why these chimpanzees did not re-innovate tool-based nut-cracking behaviour (lines 538-556), the behaviour "counts" as a CDT, an individually innovated solution, or could be either one? (This is mentioned for the other accounts.)

I’m not sure how it logically follows from the fact that other species use stones to crack nuts in the wild that social learning is not necessary for nut cracking in chimpanzees (Lines 610-613).

Specify that it is *wild* chimpanzees who go through a long trial-and-error learning process (line 626)?

---

## Round 0.2 · Minor Revisions

Thank you for submitting your revised article for review along with your response to the reviewers.

All three reviewers who reviewed your original submission have now reviewed this version and all three recommended your article for publication. I concur with their assessment. However, two of the three reviewers still had a few outstanding items that they have requested you address. One of which, noted by reviewer 3, is the lack of the inclusion of the glossary that you mentioned in your response to the reviewers. I was also unable to access this so please ensure its inclusion with your next submission.

Given how minor the outstanding comments are, I do not think it likely that I will need to send your article out for review again after this next round of revisions. I look forward to seeing your resubmission.

·

Basic reporting

The revisions have been addressed in a thorough way and I am happy to recommend this manuscript now for publication.

Experimental design

see above

Validity of the findings

see above

Reviewer 2 ·

Basic reporting

Thank you to the authors for their detailed answers to my original questions. I find that the manuscript has been much improved and suggestions by the editor and three reviewers have been sufficiently answered and used to revise the paper. There are still a few occasions where the language suggests that the results favour one explanation over another that I think should be changed, and these are listed below with other, minor comments. Overall, I think the paper provides interesting new data in the field of the evolution of tool use/culture and recommend it for publication in PeerJ once these minor revisions have been addressed.

Minor comments:
Line 38: "...are in line" - "are" is missing

Line 53: “…AND when ecological conditions allow for it” instead?

Abstract: I feel like it might be useful to have a sentence around L48 along the lines of “..therefore, we did not find strong evidence for either hypothesis” – just to easily clarify the findings?

Line 80: “Some animal cultures may be culture dependent”….some traits?

Line 93: “….ratchet effect’ (), which underlies cumulative culture () – and is responsible for the special product of cumulative culture: CDTs” – Are CDTs a product of cumulative culture? Wouldn’t they be the basis of cumulative culture? I suggest rephrasing

Line 126: “..emphasis added), Biro…” – full stop instead of comma?

Line 152: “This would provide evidence for the view that social learning is required for nut-cracking to occur.” – even when tempered by the following sentences, I still find this statement too strong, but will leave it to the Editor’s judgement on whether it should be further changed.

Line 209 – Hypotheses instead of hypothesis

Line 504 – “nuts were/are palatable”

Line 686 – I think the sentence in parenthesis should be removed – after all the same could be said of their failure to innovate the behaviour in this study

Line 693 – Again, you could delete the ‘fully’ here – the language is still subtly pointing towards ZSL as the most likely explanation, even though the results of the study clearly do not provide support for either

There are a couple of issues with the supp materials:

- In response to my prior comment (2a in their rebuttal), the authors say they've added a supplementary table with definitions of terminology, but I don't see this. Perhaps they forgot to upload it?

- There are two supp files labelled as attention to full demos, but one of these is attention to the ghost demos I believe.

Experimental design

NA

Validity of the findings

NA

Reviewer 3 ·

Basic reporting

The majority of my comments have been addressed with precision in the revised manuscript. The Introduction lays out the competing theories more concisely, and it is now clearer in which experimental phases the target behaviours occurred. Unfortunately, I did not receive a copy of the glossary mentioned in the authors’ responses (p. 13). My outstanding concerns are as follows.

The reader gets a hint as to why this study was done/what the implications are in Lines 95-97 (“the presence or absence of CDTs in chimpanzees is of particular interest”)/Lines 178-180 (“This would support the ZLS hypothesis and would suggest that chimpanzees are capable […] of individually learning […] nut-cracking”). This gestures in the direction of answering the question “Why is it important to test these hypotheses?” It would be helpful to situate the work you have put in on this question in the context of current understandings of cultures and cognition; this question was answered eloquently in the review responses (“to determine whether the reliance on social learning to develop culture is restricted to our own species or whether it is also common to other apes”). What would it mean for the theory/field if nut-cracking is found to be in chimpanzees’ ZLS (or conversely, a CDT)? Would that make chimpanzee culture a more/less apt comparison to human culture? Would it signify a larger discontinuity in human and chimpanzee cognition than previously supposed? Or, per the review responses, would it allow us to make a particular type of inference about the last common ancestor? A simple statement in a few lines could really clear this up.

I am still confused about what a culture-dependent trait (CDT) is: in Line 78-79 it is stated that CDTs are cultural traits that rely on (cannot occur in the absence of) some form of copying social learning. In Lines 92-95, it is stated that CDTs are the special products of cumulative culture. The first description appears to be much more relaxed than the second: i.e., according to the first definition, a culture-dependent trait does not *need* to be a product of cumulative culture (as in the example of whale song), while according to the second (with CDTs as special products of cumulative culture), essentially any animal “cultural trait”, if needing to be a product of cumulative culture to qualify as a CDT, would fall short.

Lines 154-157: it feels premature to discuss the results of the study before any of the details have been provided. Perhaps part of this could be moved to the Discussion.

Experimental design

Experimental design and methods are clearly described, and points from the initial review have been addressed.

Validity of the findings

Data have been provided, and conclusions are balanced. Points from the initial review have been addressed.

Additional comments

Abstract Line 38: Sentence beginning “The latter findings in line with…” seems to be missing a word.
Abstract Line 49: Sentence beginning “We conclude”: interplay -> an interplay?
Line 113: “alongside with”: one of these words is extraneous.
Line 129: are -> is?
Line 136/140: for “migratory”/”migrating”, “dispersing” (implying not just movement but a change of group) might be a clearer word to use.
Line 292-295: This long sentence should be at least 2 sentences.
Line 326: rope -> ropes?
Line 328-329: missing a “)” somewhere.
Line 502-506: sentence seems to be missing a word.
Line 506: individual’s -> individuals’
Line 536: this title seems to be missing a word.
Line 544: sentence seems to be missing a word.
Lines 629-630, 641, 645: Check spellings.
Can you rephrase the sentence in Lines 710-712? Its meaning is unclear.

---

## Round 0.3 · Minor Revisions

Thank you very much for addressing all the reviewers’ final comments. I do not feel the need to send this article back out to the reviewers and I would be happy to accept it for publication in PeerJ. However, before I can do so, I noticed a couple of issues with your supplemental materials that I must ask you to correct.

In your methods you state “The section of the questionnaire relevant to this study can be found in S1.” But I do not see this questionnaire provided in the supplementary materials.

The supplementary materials I can access are:
1. Analyses nut cracking
2. Attention full demonstration
3. Reliability nut cracking
4. Reliability coding
5. Attention ghost
6. S1 glossary

Therefore, can you please provide the questionnaire that you reference in the text or remove this statement from the body of your text.

Furthermore, please can you also make reference to your glossary (provided in supplementary materials) in the body of your article i.e. somewhere in your introduction when you present your terminology (sorry if I missed that reference, but I did not see reference to it in your article).

Additionally, to aid the reader, can you please add the age-at-testing of each subject to Table 1 in addition to their date of birth.

---

## Round 0.4 · accepted · Accept

Thank you for attending to the few outstanding questions I had. It is my pleasure to accept your article for publication in PeerJ.